# Early Assessment of Neoadjuvant Chemotherapy Response Using Multiparametric Magnetic Resonance Imaging in Luminal B-like Subtype of Breast Cancer Patients: A Single-Center Prospective Study

**DOI:** 10.3390/diagnostics13040694

**Published:** 2023-02-12

**Authors:** Lucija Kovacevic, Marko Petrovecki, Lea Korsa, Zlatko Marusic, Ivo Dumic-Cule, Maja Prutki

**Affiliations:** 1Clinical Department of Diagnostic and Interventional Radiology, University Hospital Centre Zagreb, Kispaticeva 12, 10000 Zagreb, Croatia; 2Clinical Department of Pathology and Cytology, University Hospital Centre Zagreb, Kispaticeva 12, 10000 Zagreb, Croatia; 3University North, 104 Brigade 3, 42000 Varazdin, Croatia; 4School of Medicine, University of Zagreb, Salata 3, 10000 Zagreb, Croatia

**Keywords:** breast cancer, neoadjuvant therapy, multiparametric magnetic resonance imaging

## Abstract

This study aimed to evaluate the performance of multiparametric breast magnetic resonance imaging (mpMRI) for predicting response to neoadjuvant chemotherapy (NAC) in patients with luminal B subtype breast cancer. The prospective study included thirty-five patients treated with NAC for both early and locally advanced breast cancer of the luminal B subtype at the University Hospital Centre Zagreb between January 2015 and December 2018. All patients underwent breast mpMRI before and after two cycles of NAC. Evaluation of mpMRI examinations included analysis of both morphological (shape, margins, and pattern of enhancement) and kinetic characteristics (initial signal increase and post-initial behavior of the time-signal intensity curve), which were additionally interpreted with a Göttingen score (GS). Histopathological analysis of surgical specimens included grading the tumor response based on the residual cancer burden (RCB) grading system and revealed 29 NAC responders (RCB-0 (pCR), I, II) and 6 NAC non-responders (RCB-III). Changes in GS were compared with RCB classes. A lack of GS decrease after the second cycle of NAC is associated with RCB class and non-responders to NAC.

## 1. Introduction

Breast cancer is the most commonly diagnosed cancer in the female population [1]. It is a highly heterogeneous group of tumors classified into several molecular subtypes that differ in clinical presentation, prognosis, and response to treatment [2,3]. Due to logistical and financial reasons, surrogate subtype definitions based on immunohistochemical analysis of breast cancer tissue specimens are mainly used in clinical practice. Surrogate subtypes of breast cancer are luminal A-like, luminal B-like, HER2-positive, and triple-negative [4,5,6].

In recent decades, there has been a shift in breast cancer treatment toward increased use of preoperative systemic chemotherapy, also known as neoadjuvant chemotherapy (NAC). NAC offers several additional advantages over established adjuvant therapy while maintaining similar efficacy [7,8,9,10,11,12]. One of the main advantages of NAC over adjuvant chemotherapy is that it allows monitoring of response to treatment in individual patients. The standard for assessing response to NAC is histopathological tissue analysis, as it provides important prognostic information [13]. Although pathologic complete response (pCR) is associated with improved long-term outcomes, binary classification of NAC response lacked prognostic information in a heterogeneous group of patients with different amounts of residual disease. The residual cancer burden (RCB) method was developed to overcome the binary classification problem [14]. The RCB method divided the group of non-PCR patients into three groups based on residual disease.

Recently, it has been recognized that the RCB score and RCB class are independently prognostic in all subtypes of breast cancer and have been suggested as the preferred prognostic score for guiding treatment decisions after NAC [15,16,17].

Breast cancer heterogeneity is the major challenge in the optimal treatment of breast cancer patients because different histological and immunophenotypic subtypes of breast cancer respond differently to NAC. In patients with luminal B-like breast cancer, the use of NAC remains controversial, because such tumors rarely achieve pCR after NAC [18], and patients are exposed to both short-term and long-term sequelae of chemotherapy. Nowadays, several less toxic treatment options are available for this subgroup of patients. Furthermore, approximately 5% of breast cancer patients experience disease progression during NAC, and a proportion of patients experience NAC toxicity and delays in more effective treatment [19]. Therefore, early assessment of response in patients with luminal B-like breast cancer is crucial to allow changes in systemic therapy and to expedite definitive surgery in patients who show early disease progression during NAC.

To avoid toxicities and delays in more effective treatment for this subgroup of breast cancer patients, it is crucial to develop new biomarkers that would allow the optimization of treatment at an early stage.

Since both clinicopathological factors and imaging findings are routinely assessed before and during NAC, their potential as predictive biomarkers of response to NAC in breast cancer has been extensively studied. Breast MRI has been shown to be superior to clinical examination, mammography, and breast ultrasound in both early response prediction and presurgical evaluation [20,21,22,23,24,25]. Multiparametric MRI can detect changes in breast cancer morphology and function that occur as a result of NAC application and therefore predict response to NAC [26,27]. Multiparametric MRI interpreted with a Göttingen score (GS) utilizes both morphological (shape, margins, and pattern of enhancement) and kinetic characteristics (initial signal increase and post-initial behavior of the time-signal intensity curve) of breast cancer [28,29,30]. The aim of this study is to determine whether changes in GS, which occur early during NAC, are predictive of NAC response.

## 2. Materials and Methods

### 2.1. Patient Characteristics

The institutional review board approved the study, and informed consent was obtained from all patients before enrollment. This single-center prospective study included 35 female patients diagnosed with breast cancer of the luminal B-like subtype at the University Hospital Centre Zagreb between January 2015 and December 2018. All patients underwent breast mpMRI at the UHC Zagreb before any treatment, after the second cycle of NAC, and underwent NAC with known outcomes in terms of RCB class (as determined at the time of surgery). All patients received doxorubicin-cyclophosphamide (AC) followed by weekly paclitaxel as neoadjuvant chemotherapy (NAC) for breast cancer. Additionally, according to the standard protocol, patients with HER2-positive disease received a Trastuzumab or a combination of Pertuzumab and Trastuzumab. Patients with incomplete tissue analysis and poor image quality before treatment and after the second cycle of NAC were not included. The patient selection process is summarized in Figure 1.

### 2.2. Multiparametric Magnetic Resonance Imaging Aquisition Protocol

The study included images acquired at the Department of Diagnostic and Interventional Radiology at University Hospital Centre Zagreb by 1.5 T MR unit (Avanto, Siemens, Erlangen, Germany) using a dedicated breast coil, with the patient in a prone position with both hands flat on the sides of the body. The bilateral breasts were fully exposed and naturally fell into the breast phased-array coils. The complete examination included the following sequences: (1) Turbo Inversion Recovery Magnitude axial sequence (Repetition Time (TR) = 5600.0, Echo Time (TE) = 59.0, Bandwidth 252.0, Matrix 1.0 × 0.7 × 1.0, Thickness = 4.0, Interval = 0.8 mm, Field of View (FOV) = 340 mm, Number of Excitations (NEX) = 2); (2) T2 weighted nonFatSat axial sequence (TR = 5000.0, TE = 94.0, Matrix 0.8 × 0.8 × 4.0, Thickness = 4.0, Interval = 0.4 mm, FOV = 370 mm, NEX = 2); (3) 3D T1 weighted FatSat axial sequence (TR = 4.06, TE = 1.65, FA = 10.0, Matrix1.0 × 0.8 × 1.5, Thickness 1.0, Interval 0.0, FOV = 320 mm, NEX = 1) before and five times after intravenous administration of 0.1 mmol/kg body weight of gadoterate meglumine (Dotarem^®^, Guerbet, Princeton, NJ, USA) into the antecubital vein using an automatic injection system at a flow rate of 3.5 mL/s followed by a flush of 20 mL of saline solution. A total of 6 phases were scanned, and the first phase involved plain scanning. The injected contrast agent began to enhance the scan for a total of 5 phases. (4) DWI axial sequences (TR = 8412.0, TE = 129.0, Matrix 1.9 × 1.9 × 5.0, Thickness 5.0, Interval 0.5, FOV = 370 mm, NEX = 2) with b values 50, 750, 1000. An ADC map was automatically constructed in a commercially available workstation, the values were expressed in mm^2^/s; (5) MR—spectroscopy (TR = 1500.0, TE = 100.0, Voxel size 10.0 × 10.0 × 10.0 mm, NEX 128.0, Matrix 0.8 × 0.8 × 4.0, Thickness = 4.0, Interval = 0.4 mm, FOV = 370 mm, NEX = 2, vector size 1024.0. The presence or absence of choline in the breast spectra was determined by the detection of the peak at 3.2 parts per million (ppm) within the lesion.

### 2.3. Image Analysis

The postprocessing evaluation of all breast MRI examinations included image subtraction of the dynamic images and maximum intensity projections (MIPs) of subtracted data to better identify enhancing lesions and time-enhancement curves for suspicious lesions. Initially, all mpMRI examinations were evaluated separately by two radiologists (with 5 and 10 years of experience in breast MRI) who were blinded to other information. After the initial evaluation, experts accepted concordant results as final and discussed any discrepant results until a consensus was reached. The abnormal enhancement was classified as mass or non-mass enhancement. The mass lesion morphology characteristics, including shape (oval, round, irregular), margin (circumscribed, not circumscribed (irregular, speculated)) and internal enhancement pattern (homogeneous, heterogeneous, rim enhancement), were evaluated. For each mass lesion, initial enhancement and kinetic curve analysis were performed. Signal intensities were measured on precontrast and each postcontrast series by placing the smallest possible (5 pixels) operator-defined region of interest (ROI) at the most enhanced part of the lesion on the first postcontrast image. Initial enhancement was calculated using the (Signal_postinitial_ – Signal_initial_)/Signal_initial_ × 100% formula. Three types of kinetic curves were observed according to post-initial enhancement including Type I (persistent), Type II (plateau), and type III (washout). Tumor size was estimated by measuring the largest tumor diameter on the axial MIP images. According to the Göttingen score, points were assigned to each of the five parameters (shape, margins, internal enhancement pattern, initial enhancement, and type of kinetic curve) (Table 1), and the total score was calculated [28].

### 2.4. Histopathological Analysis

Histopathological analysis of the initial breast biopsy or surgical excision included staining formalin-fixed, paraffin-embedded 5-μm-thick tissue sections representative of the tumor and assessment of histological type, histological grade, and immunohistochemical (IHC) analysis of estrogen receptor (ER), progesterone receptor (PR), HER2 and Ki-67 status. The expression of ER, PR, HER2, and Ki-67 was assessed by immunostaining with commercially available antibodies (all Ventana, Tucson, AZ, USA). The ER and/or PR status was positive when at least 1% of the tumor cell nuclei showed staining for ER or PR, according to the Breast Biomarker Reporting guidelines of the College of American Pathologists (CAP). The HER2 expression was semiquantitatively assessed by the intensity and percentage of staining of tumor cells, and scored on a scale of 0–3+. Scores of 0 and 1+ were categorized as negative, 2+ as equivocal, and 3+ as positive. Specimens scored as equivocal were subsequently retested by Silver In Situ Hybridization (SISH). The cut-off point for the positive expression of Ki-67 was 20%. Surrogate definitions based on IHC analysis of breast cancer tissue were used to determine the breast cancer subtypes, and only patients with luminal B-like subtype were included in this study. Histopathological analysis of surgical samples additionally included evaluation of tumor size, lymph node involvement, and assessment of treatment response after the completion of NAC using the Residual Cancer Burden (RCB) grading system. Tumor response was classified into four classes (RCB-0 (pCR), RCB-I, RCB-II, RCB-III) based on the increasing extent of residual disease. Based on RCB classes, patients were divided into two groups, with RCB classes 0 (pCR), I, and II being the group named Responders and others with RCB class III being the group non-Responders.

### 2.5. Statistical Analysis

The Wilcoxon test was used to analyze mean values of tumor size and Göttingen score before and during NAC because of the small sample size and data that does not follow a normal distribution. Four predictor variables showing changes in tumor characteristics occurring during the first two cycles of NAC were calculated for each patient. Those predictor variables include both absolute (tumor size difference, Göttingen score difference) and relative (proportion of tumor size difference and the proportion of Göttingen score difference) changes in tumor characteristics occurring early during NAC treatment. Univariate logistic regression analysis was performed to determine the association between NAC response (groups Responders vs. non-Responders) and each of the four predictor variables: tumor size difference, the proportion of tumor size difference, Göttingen score difference, and the proportion of Göttingen score difference. Variables found to be significant (*p*-value less than 0.1) by univariate logistic regression analysis were included in the multivariate logistic regression model. Multivariate logistic regression analysis was performed to determine the independent predictors of NAC response. Here, variables with a *p*-value less than 0.05 were considered significant prediction factors. The calculations were performed using MedCalc (version 12.7.3.0—Windows XP/Vista/7/8, MedCalc Software bvba, Ostend, Belgium; http://www.medcalc.org; accessed on 15 October 2022. 1993–2013).

## 3. Results

A total of 35 Caucasian female breast cancer patients were included in the study. The mean age was 55 years (range: 30–73 years). Most women enrolled in the study had early-stage tumors (≤IIA), with 60% of all patients having node-positive breast cancer. Estrogen and progesterone receptors were positive in 35 (100%) and 29 (83%) cases, respectively. All breast cancers analyzed in the study were of the luminal B-like surrogate subtype, with 13 (37%) being HER2-positive and 22 (63%) being HER2-negative. The predominant histological subtype was invasive carcinoma of no special type (100%). After the completion of NAC, most patients 29 (83%) achieved RCB classes 0, I and II and were assigned to the group responders to NAC. Detailed patient characteristics are shown in Table 2.

All 35 (100%) breast cancers presented as mass lesions, and there were no non-mass enhancement lesions. The mean tumor size before the application of NAC was 35 mm (range 15–95 mm), and the mean tumor size after the second cycle of NAC was 25 mm (11–93 mm). On pretreatment MRI, the shape was round or oval in two (6%) and irregular in 33 (94%) tumors. Both tumors with round or oval shapes had circumscribed margins (6%), while margins were not-circumscribed in the remaining 33 (94%) tumors. After the second cycle of NAC, only one tumor presented with a round shape, while the remaining 34 (97%) tumors presented with an irregular shape. The internal enhancement pattern was rim enhancement in 17 (49%) cases before the onset of NAC treatment and 11 (31%) cases after the second cycle of NAC. Heterogeneous enhancement was observed in 18 (51%) patients before the onset of NAC treatment and 24 (69%) patients after the second cycle of NAC. None of the tumors presented with a homogenous enhancement pattern both before and after the second cycle of NAC application. Initial enhancement observed on pretreatment mpMRI was less than 50% in three (9%) cases, 50–100% in 20 (57%) cases, and >100% in 12 (34%) cases. After the second cycle of NAC, initial enhancement was less than 50% in 12 (34%) cases, 50–100% in 19 (54%) cases, and >100% in four (12%) cases. On pretreatment mpMRI, the kinetic curve types were plateau in 13 (37%) tumors and wash-out in 22 (63%) tumors, while during NAC, a persistent curve was observed in nine (26%) cases, plateau in 18 (51%) cases, and wash-out in eight (23%) cases. The mean Göttingen score before NAC was 6 (range 4–8), and after the second cycle of NAC it was 5 (range 3–8). The detailed data are shown in Table 3.

The mean value of tumor size change between MRI scans taken before and after two cycles of NAC was 5 mm (range −3–47 mm), while the mean value of percentage of the tumor reduction was 14% (range −12–70%). The mean value of GS change between MRI scans taken before and after two cycles of NAC was 1 (range 0–4), while the mean value of the percentage of GS decrease was 20% (range 0–57%). Both characteristics showed a significant difference in mean values between pretreatment MRI and MRI acquired after 2nd cycle of NAC. The data are listed in Table 4.

Univariate logistic regression analysis was performed to identify the predictor variables that have a significant impact on NAC response (groups Responders vs. non-Responders). Predictor variables used in univariate logistic regression represent changes in tumor characteristics observed on breast MRI that occur during the first two cycles of NAC and include: tumor size difference, the proportion of tumor size difference, Göttingen score difference, and the proportion of Göttingen score difference. Both tumor size difference and the proportion of tumor size difference are not associated with the final NAC response. Univariate logistic regression showed that the Göttingen score difference and the proportion of Göttingen score difference are significantly associated with NAC response. Variables found to be significant (*p*-value less than 0.1) by univariate regression were included in the multivariate logistic regression model to identify significant independent predictors. No significant independent predictors of NAC response were found by multivariate analysis. Univariate and multivariate analyses of the impact of differences in tumor characteristics before and after the second cycle of NAC are listed in Table 5.

## 4. Discussion

In terms of perioperative chemotherapy, the treatment of breast cancer is focused on NAC due to its several benefits over classic adjuvant therapy. However, the use of NAC in the subgroup of patients who have luminal B-like breast cancer is still controversial, as these tumors usually do not achieve pCR following NAC. Therefore, early assessment of response to NAC for patients with luminal B-like breast cancer is crucial.

The main advantage of NAC over adjuvant therapy is the opportunity to assess response to NAC at an early stage of NAC as a predictor of pathologic response and therefore enable treatment modification to increase the probability of pCR, tumor volume reduction, and treatment tolerability [21]. In the NAC setting, mpMRI is the best imaging method for assessing early response to the NAC, since it captures not only morphological but also functional changes that may occur before morphological changes [21]. Different tools have recently been tested for the predictability of breast cancer response after NAC. Changes in lesion size, volume, and enhancement pattern are used to predict response to NAC [27]. The predictive effect of the change in tumor size during NAC on the final outcome has been proven by several studies. After two cycles of NAC, it is possible to predict the final response to NAC by the reduction in the largest tumor diameter, tumor reduction in two measurement dimensions, and tumor volume reduction [27,31,32,33,34]. In our study, the largest tumor diameter change failed to predict the final NAC response. Background parenchymal enhancement parameter in MRI was recently suggested as a possible predictive factor in response to NAC in breast cancer [35]. Synthetic MRI was successfully utilized to predict which breast cancer would achieve a pCR after NAC administration [36]. The main limitation of this study was the sample size, which is comparable to our study and needs further verification.

A novel quantitative approach to medical imaging called radiomics has the potential for early NAC response prediction [37]. Most of the published studies on radiomic models based on radiomic features extracted from the tumoral region of breast MRI have shown great predictive potential and yielded an area under the curve (AUC) up to 0.94 [38,39]. Moreover, better predictive performances are obtained if MRI-based radiomic features from the tumoral region are combined with radiomic features from the peritumoral region and clinical parameters [40].

However, promising radiomics approaches are still not widely available in daily clinical practice, and one of the issues is the lack of knowledge of its basic concepts among radiologists [41]. Therefore, although a growing number of studies in the field of early-response prediction to NAC is in the domain of radiomics, this study focused on the potential of GS for NAC response prediction. GS is a simple MRI interpretation tool that combines morphological and functional parameters. Since it does not require additional competencies among radiologists, it is easier to incorporate it into radiologists’ everyday workflow. As a tool for MRI interpretation, GS has previously been shown to be a significant, independent predictor of a higher histological grade of invasive ductal carcinoma [42].

To avoid waiting for histopathology to test NAC response, we aimed to assess whether changes in GS that occur at an early stage of NAC treatment are predictive of NAC response. To the best of our knowledge, according to the available literature, the potential of GS in such prediction has not been suggested so far. Our study revealed that lack of decrease in GS after the second cycle of NAC was associated with RCB after surgery and non-responders to NAC.

There are several limitations to this study, including the small sample (a total of 35 patients) and the small proportion of patients without response to NAC (only six patients had RCB-III), which makes linear regression results challenging to interpret. The odds ratios are very low, with a wide range of 95% confidence intervals. A larger sample size could have identified some statistically significant predictors of NAC response. Furthermore, inter-observer variability could not be analyzed, since the interpretation of MRI examinations was performed by two radiologists on the basis of consensus.

Collectively, mpMRI has the potential to become a valuable diagnostic tool for assessing early response to NAC in patients with luminal B-like breast cancer, subsequently allowing changes in systemic therapy and acceleration of definitive surgery in patients who progress early during NAC.

## Figures and Tables

**Figure 1 diagnostics-13-00694-f001:**
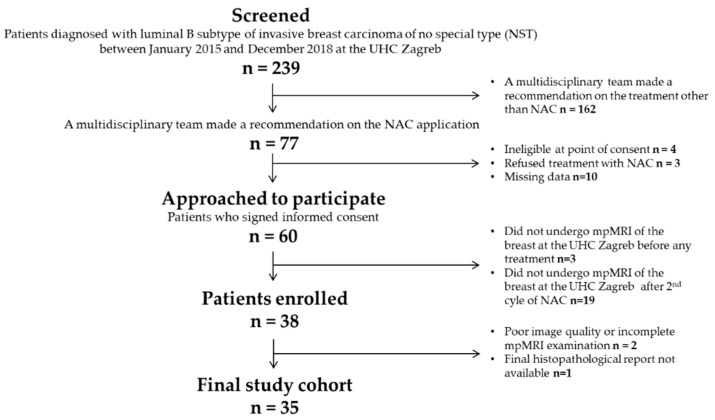
Flow chart of the patient selection process.

**Table 1 diagnostics-13-00694-t001:** Göttingen score (GS) for the evaluation of mass lesions [28].

Criteria	Points		
	0	1	2
Shape	Round, oval	Irregular	-
Margins	Circumscribed	Not-circumscribed	-
Enhancement pattern	Homogeneous	Heterogeneous	Rim enhancement
S_initial_	<50%	50–100%	>100%
S_postinitial_	Continous increase	Plateau	Wash-out

**Table 2 diagnostics-13-00694-t002:** Patient characteristics.

Patient Characteristics	
**Age**	
Value (mean, range)	55 (30–73)
**Clinical Staging**	
I (N, %)	13 (37%)
IIA (N, %)	12 (34%)
IIB (N, %)	8 (23%)
IIIA (N, %)	2 (6%)
**Histological type**	
Non-special type invasive carcinoma	35 (100%)
Lobular infiltrating carcinoma	-
Other	-
**Nuclear grade**	
1	-
2	18 (52%)
3	17 (48%)
**Estrogen receptor status**	
Positive (N, %)	35 (100%)
Negative (N, %)	-
**Progesterone receptor status**	
Positive (N, %)	29 (83%)
Negative (N, %)	6 (17%)
**HER2 receptor status**	
Positive (N, %)	13 (37%)
Negative (N, %)	22 (63%)
**Ki-67**	
Value (mean, range)	41% (12–87%)
**Lymph node status**	
Positive (N, %)	21 (60%)
Negative (N, %)	14 (40%)
**Tumor response after the completion of NAC based on the Residual Cancer Burden (RCB) grading system**	
Responders (N, %)	29 (83%)
non-Responders (N, %)	6 (17%)

**Table 3 diagnostics-13-00694-t003:** Imaging features of breast cancer before and after the second cycle of NAC.

Imaging Features	Before Application of NAC	After 2nd Cycle of NAC
**Tumor size**		
Value (mean, range)	35 mm (15–95 mm)	25 mm (11–93 mm)
**Tumor shape**		
Round, oval (N, %)	2 (6%)	1 (3%)
Irregular (N, %)	33 (94%)	34 (97%)
**Tumor margins**		
Circumscribed (N, %)	2 (6%)	5 (14%)
Not-circumscribed (N, %)	33 (94%)	30 (86%)
**Tumor enhancement pattern**		
Homogeneous (N, %)	-	-
Heterogeneous (N, %)	18 (51%)	24 (69%)
Rim enhancement (N, %)	17 (49%)	11 (31%)
**S_initial_**		
<50% (N, %)	3 (9%)	12 (34%)
50–100% (N, %)	20 (57%)	19 (54%)
>100% (N, %)	12 (34%)	4 (12%)
**S_postinitial_**		
Continous increase (N, %)	-	9 (26%)
Plateau (N, %)	13 (37%)	18 (51%)
Wash-out (N, %)	22 (63%)	8 (23%)
**Göttingen score**		
Value (mean, range)	6 (4–8)	5 (3–8)

**Table 4 diagnostics-13-00694-t004:** Differences and percentages of differences in tumor characteristics before and after second cycle of NAC.

Tumor Characteristics	Difference in Tumor Characteristics before and after Two Cycles of NAC (Mean Value, Range)	Proportion of the Difference from the Initial Value (Mean VALUE, Range)	Wilcoxon Test *p* (Z)
Tumor size	5 mm (−3–47 mm)	14% (−12–70%)	*p* < 0.001 (Z = 4.78)
Göttingen score difference	1 (0–4)	20% (0–50%)	*p* < 0.001 (Z = 4.62)

**Table 5 diagnostics-13-00694-t005:** Univariate and multivariate analyses of the impact of differences in tumor characteristics before and after second cycle of NAC.

Analysis	OR	95% CI	*p*
**Univariate regression**			
Tumor size difference	0.84	0.65–1.01	0.131
The proportion of tumor size difference	0.004	<0.001–3.12	0.087
Göttingen score difference	0.09	0.02–0.65	0.016
The proportion of Göttingen score difference	0.001	<0.001–0.04	0.012
**Multivariate regression**			
The proportion of tumor size difference	0.001	<0.001–7285.65	0.151
Göttingen score difference	0.003	<0.001–3.01	0.1

## Data Availability

Data used for analysis is contained within the article.

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
