# Peer review of "Early Assessment of Neoadjuvant Chemotherapy Response Using Multiparametric Magnetic Resonance Imaging in Luminal B-like Subtype of Breast Cancer Patients: A Single-Center Prospective Study"

_diagnostics, 2023, doi:10.3390/diagnostics13040694_

Round 1

Reviewer 1 Report

I am grateful for the opportunity to review manuscript ID: diagnostics-2156595, entitled “Early assessment of neoadjuvant chemotherapy response using multiparametric magnetic resonance imaging in luminal B-like subtype of breast cancer patients: A single center prospective study”

The authors present a prospective evaluation of multiparametric magnetic resonance imaging interpreted with Göttingen scoring system for NAC response in monocentric cohort of patients with a luminal B-type breast cancer.

The manuscript is well structured, straightforward narrative review on the topic. The tables and the citations are appropriate. I would recommend its publication after the authors address the comments below.

Some minor comments:

Methods 2.3: Please consider reporting your decision protocol for cases of disagreement between the two observers.

Methods 2.4: RCB groups are usually denoted by Roman numerals. Clarifying RCB-0 being equivalent to pCR could assist a better understanding of the readers.

Would you please consider reporting which substances were given in terms of NAC?

Line 167: Were all 35 patients female, as it could be assumed?

Line 207 & 220: (Tables 4 & 5): Please use appropriate decimal separators.

Please revise lines 252-253.

Author Response

We thank this reviewer for the constructive criticism which has significantly improved the quality of our revised manuscript. In the following points we explain how we addressed the minor comments suggested by the reviewer.

  • Methods 2.3: Please consider reporting your decision protocol for cases of disagreement between the two observers.

Based on the reviewer's comments, we included the following paragraph in the Methods 2.3. section: “Initially, all mpMRI examinations were evaluated separately by two radiologists (with 5 and 10 years of experience in breast MRI) who were blinded to other information. After the initial evaluation, experts accepted concordant results as final and discussed any discrepant results until a consensus was reached.”

  • Methods 2.4: RCB groups are usually denoted by Roman numerals. Clarifying RCB-0 being equivalent to pCR could assist a better understanding of the readers.

We thank the reviewer for pointing this out. According to your suggestions, we made changes in the manuscript in the following sections: Abstract; 2.4. Pathoistological analysis; 4. Discussion.

  • Would you please consider reporting which substances were given in terms of NAC?

We thank the reviewer for pointing this out. Based on the reviewer’s comments, we included the following paragraph in Methods 2.1. Patient characteristics section: “All patients received doxorubicin-cyclophosphamide (AC) followed by weekly paclitaxel as neoadjuvant chemotherapy (NAC) for breast cancer. Additionally, according to the standard protocol, patients with HER2-positive disease received a Trastuzumab or a combination of Pertuzumab and Trastuzumab.”

  • Line 167: Were all 35 patients female, as it could be assumed?

We thank the reviewer for pointing this out. All patients were women, and we clarified this in our manuscript's revised version in the following sections 2. Methods 2.1. Patient characteristics, and 3. Results.

  • Line 207 & 220: (Tables 4 & 5): Please use appropriate decimal separators.

We thank the reviewer for pointing this out. According to your suggestions, we made changes in the manuscript.

  • Please revise lines 252-253.

We thank the reviewer for pointing this out. We have removed a part of the text ‘’For research articles with several authors, a short paragraph specifying their individual contributions must be provided. The following statements should be used.’’

We believe that the current version has gained both clarity and readability and will therefore be suitable for publishing in your distinguished Journal and of interest to the broad audience.

Ivo Dumic-Cule, MD, PhD

Reviewer 2 Report

The manuscript entitled ‘Early assessment of neoadjuvant chemotherapy response using multiparametric magnetic resonance imaging in luminal B-like subtype of breast cancer patients: A single center prospective study’ investigated the treatment response of luminal B subtypes of breast cancers. The topic is interesting. However, the current manuscript is not suitable to be published. The issues should be addressed. 1. Is the sample size enough for further analysis? 2. The citations were not suitable. Too old publications to do further analysis. 3. No other imaging results for comparison. The comparison between before and after NAC is not appropriate, which is supposed to be reduced accordingly. 4. Why did the authors choose luminal B subtypes? How about the comparison between different subtypes? 5. In different clinicopathological parameters, how to set the cut-off?

Author Response

We are glad to know that this reviewer finds our research topic interesting. We also thank the reviewer for expressing his concerns, which helped us to improve our work. Our point-by-point responses to the reviewers' comments are listed below.

  • Is the sample size enough for further analysis?

We thank the reviewer for pointing this out. Since this was a prospective study, we had a pre-defined period for patient enrollment which resulted in the inclusion of 35 patients. Given that the number of patients included in the study could not be influenced, we performed a post-hoc power analysis to support our conclusions. After completing the post-hoc power analysis, we determined that the effect size is quite large, d=1.37, which for our sample, yields a statistical power over 0.82. This suggests that our finding of statistically significant differences between the groups will probably be reproducible in future studies.

  • The citations were not suitable. Too old publications to do further analysis.

We thank the reviewer for pointing this out. According to your suggestions, in the revised version of our manuscript, we placed our research more in the context of recent studies on the capabilities of medical imaging in predicting NAC response. Accordingly, we changed the following sections of our manuscript: Introduction, Discussion, and References.

  • No other imaging results for comparison. The comparison between before and after NAC is not appropriate, which is supposed to be reduced accordingly.

We thank the reviewer for pointing this out. However, this study aimed to predict neoadjuvant therapy response based on changes in Göttingen score (GS) that occur early during neoadjuvant treatment. We used breast MRI examinations acquired before NAC and after the 2nd cycle of NAC (complete NAC has 16 cycles) to search for changes predictive of NAC response. We did not compare MRI examinations acquired before and after NAC since this research was to allow for early treatment changes in the subgroup of breast cancer patients that rarely achieve favorable outcomes and are exposed to chemotherapy toxicities.

NAC response was assessed during a histopathological analysis of post-NAC surgical specimens since it is considered a gold standard for NAC response assessment in clinical practice.

Although a growing number of studies use the radiomics approach of breast MRI analysis for NAC response prediction, we wanted to explore the potential of GS (a simple tool for MRI interpretation) for NAC response prediction since it doesn't require new skills and, therefore, it is easier to incorporate it in everyday radiologists' workflow. We did not compare other imaging modalities as it is known that breast mpMRI is the most accurate modality to demonstrate, monitor, and predict response to NAC therapy. Furthermore, lesion diameter, volume, and enhancement changes are already used to predict response to NAC. However, despite the reported predictive power of those features, none of these studies have used a simple scoring system like GS that combines both morphological and functional parameters to assess predictive power for NAC response prediction.

  • Why did the authors choose luminal B subtypes? How about the comparison between different subtypes?

We thank the reviewer for pointing this out. Due to the prospective design of this study, we had predefined inclusion criteria. When designing the study, we decided to include only patients with luminal B-like subtype breast cancer in this study for the reasons listed below.

  1. Luminal B-like subtype of breast cancer is the most common subtype of breast cancer treated with NAC at UHC Zagreb
  2. Luminal B-like subtype of breast cancer rarely achieves favorable NAC response
  3. Patients with luminal B-like subtype of breast cancer are exposed to NAC toxicity and reduced quality of life, although they can be treated with other methods of preoperative systemic (e.g., hormone therapy) and local treatment, which may be less toxic and less disruptive to quality of life.

Therefore, as has been already stated, this study aimed to detect patients who will not respond favorably to NAC to allow for timely treatment change. Although they do not achieve a good response while treated with NAC, they represent the majority of breast cancer patients treated with NAC in our institution. We could not compare different subtypes of breast cancer since during the time of patient inclusion in this study. We had only three patients with luminal A subtype of breast cancer, four patients with HER2-positive breast cancer, and five with triple-negative breast cancer treated with NAC and had pretreatment mpMRI acquired at the UHC Zagreb.

  • In different clinicopathological parameters, how to set the cut-off?

We thank the reviewer for pointing this out. However, to set a boundary in different clinicopathological parameters, we should conduct more extensive studies that include patients with other subtypes of breast cancer.

This study aimed to allow for better decision-making in treating patients with luminal B-like subtype of breast cancer since the use of NAC in that group of breast cancer patients is still controversial due to poor NAC response.

In contrast, NAC is the preferred treatment for early and locally advanced breast cancer of the HER2-positive and triple-negative subtypes and is not recommended as a treatment of choice for patients with luminal A subtype breast cancer.

Thus, we sought to identify patients with a luminal B-like subtype of breast cancer in which the benefits of NAC use would outweigh the cons, such as toxicity and possible delays in more effective treatment. However, further research must be done to set the cut-off in different clinicopathological parameters.

We believe that the current version has gained both clarity and readability and will therefore be suitable for publishing in your distinguished Journal and of interest to the broad audience.

Sincerely Yours,

Ivo Dumic-Cule, MD, PhD
